# ENHANCING PERSONA FOLLOWING AT DECODING TIME VIA DYNAMIC IMPORTANCE ESTIMATION FOR ROLE-PLAYING AGENTS

**Yuxin Liu**[1]  **Mingye Zhu**[1]  **Siyuan Liu**[1]  **Bo Hu**[1]  **Lei Zhang**[1*]
[1]University of Science and Technology of China
`{liuyuxin1010, mingyezhu, liusiyuan0524}@mail.ustc.edu.cn`
`{hubo, leizh23}@ustc.edu.cn`

## ABSTRACT

The utility of Role-Playing Language Agents in sociological research is growing alongside the adoption of Large Language Models. For realism in social simulation, these agents must adhere to their personas defined by character profiles, yet existing strategies—static prompt engineering or costly fine-tuning—fail to adapt personas to dynamic scenarios. Psychological theories, such as the Cognitive-Affective Personality Systems, provide a crucial explanation for this failure: a persona's influence on behavior is not static but varies with the scenarios. This context-dependence highlights the critical need for adaptive persona management. To address this gap, we propose a novel, theory-driven method that dynamically estimates context-dependent persona importance and integrates it into weighted reward-guided decoding, enabling inference-time persona following. Specifically, we introduce **P**ersona **D**ynamic **D**ecoding (**PDD**) framework that consists of two key components: (1) Persona Importance Estimation (PIE) module, which dynamically quantifies the contextual importance of persona attributes without requiring ground-truth supervision; and (2) Persona-Guided Inference-Time Alignment (PIA) paradigm, which leverages these importance scores to construct weighted multi-objective rewards and modulate generation probabilities during inference. Extensive experiments show the effectiveness of our method in utterance consistency and behavioral fidelity.

## 1 INTRODUCTION

The advancement of Large Language Models (LLMs) parallels a wider recognition of its importance to Role-Playing Language Agents (RPLAs) in sociological studies (*e.g.*, voting behavior analysis (Zhang et al., 2024), rumor diffusion dynamics (Mou et al., 2024)). To ensure real-world validity and statistical generalizability, such simulations require carefully curated, heterogeneous persona profiles representing diverse sociodemographic segments. Critically, psychological research on persona dynamics (Sherman et al., 2015; Green et al., 2019), including the Cognitive-Affective Personality Systems (CAPS), establishes that human behavior emerges from scenario-specific attribute activation. Mirroring this, LLMs should dynamically identify context-dependent personas and adhere to predefined profile across varying contextual scenarios—a capability we term *Persona Following*.

Existing techniques for persona following primarily fall into two categories: non-parametric learning and parametric training. Representative non-parametric approaches include Direct Prompting, In-Context Learning (ICL) (Dong et al., 2024), and Retrieval-Augmented Generation (RAG) (Lewis et al., 2020). These methods predominantly rely on prompt engineering, where carefully designed textual prompts inject persona-specific descriptors into the model to steer its outputs toward predefined profiles. However, as their core mechanism hinges on semantic recognition of prompt text, models often fail to achieve profound comprehension of the persona attributes embedded within inputs. Consequently, they cannot dynamically adapt behavioral patterns or preference expressions

---

*Lei Zhang is the Corresponding author.

according to specific scenarios. This fundamental limitation impedes genuine dynamic persona following during agent interactions. To address these issues, parametric approaches (Shao et al., 2023; Zhou et al., 2024) optimize models via Supervised Fine-Tuning (SFT) or Low-Rank Adaptation (LoRA) on large-scale curated corpora (*e.g.*, character dialogues (Xu et al., 2024)) to enhance persona following. However, such methods demand substantial computational resources and annotated data. This challenge is exacerbated in social simulations, where diverse roles with complex personalities make dataset curation highly resource-intensive. In summary, existing methods exhibit two core limitations: (1) Deficient dynamic adaptability: Failure to recognize context-dependent persona across scenarios, and (2) Heavy data dependency: Extreme difficulty in curating behavioral data covering diverse scenarios.

To address existing limitations, we propose **Persona Dynamic Decoding (PDD)**, an inference-time persona following framework, which dynamically adapts persona importance to varying scenarios and guides generation without any fine-tuning. PDD comprises two key components: Persona Importance Estimation (PIE), which adaptively quantifies persona importance across different scenarios, and Persona-Guided Inference-Time Alignment (PIA), which incorporates the importance scores estimated by PIE into a reward function that modulates the model's output, thereby aligning it with the target persona during inference. Firstly, PIE quantifies the influence of each persona attribute by assessing the Conditional Mutual Information (CMI) of this attribute and model output. Specifically, the CMI is approximated using only inference-time log probabilities, thus eliminating the reliance on ground-truth supervision—an impractical requirement in multi-scenario and multi-persona role-playing tasks. Furthermore, we theoretically show that, under mild assumptions, model-generated outputs provide a reliable basis for deriving importance rankings, even in the absence of gold-standard responses. This enables our approach to dynamically identify context-dependent personas, facilitating adaptive persona following across diverse scenarios. Building upon PIE, we further propose PIA paradigm, which decomposes salient persona attributes from agent profiles into composite persona-following objectives. Specifically, the persona importance obtained from PIE are used to weight each attribute's step-wise reward, forming an overall multi-objective reward function. This reward dynamically modulates token-level generation probabilities during decoding, steering the model toward outputs that more accurately reflect the most relevant persona attributes. Unlike prior approaches, PIA functions at inference time and is capable of dynamically adapting to diverse scenarios.

> 📈 Contributions
>
> ***Concepts***: Departing from conventional RPLAs that rely on static persona weights, this work introduces, for the first time, the principle of dynamically modulating a persona's contribution in response to varying scenarios, providing new insights for agent behavior modeling in social simulation. ***Technology***: We propose PDD, which introduces a novel PIE module to bridge scenarios and personas by dynamically quantifying the importance of different persona attributes across varying scenarios. Coupled with PIA paradigm, it integrates multi-objective reward signals and modulates generation during inference, ensuring outputs remain faithfully aligned with the predefined profiles. ***Experiments***: Extensive experiments on three role-playing benchmarks (CharacterEval, BeyondDialogue, and PERSONALITYBENCH) demonstrate PDD's superiority through comprehensive multi-dimensional evaluation.

## 2 RELATED WORK

### 2.1 ROLE-PLAYING LANGUAGE AGENTS

The development of Large Language Models (LLMs) has significantly catalyzed the rise of Role-Playing Language Agents (RPLAs) (Chen et al., 2024). Such agents can simulate a role's emotions, actions and tones, allow users to freely customize characters for interactions, attracting attentions from academics and industry (*e.g.*, Glow and Character.AI). The categories of RPLAs span from fiction characters and celebrities to daily life characters, supporting various scenarios, *e.g.*, sociological studies(Yang et al., 2024; Huang et al., 2024; Tang et al., 2025; Wang et al., 2024b). To enhance role-playing performance, existing approaches leverage either training or prompting techniques with high-quality character-specific dialogues. Early research primarily utilized LLMs'

instruction-following and knowledge acquisition capabilities through prompt engineering, incorporating specific character profiles or few-shot examples (Tang et al., 2023; Wang et al., 2024a). Recent studies MMRole (Dai et al., 2025) extends role-playing scenarios to multimodal domains, while Timechara (Ahn et al., 2024) addresses spatiotemporal hallucination issues in RPLAs through innovative prompt design. To achieve character customization through parametric training, specialized LLMs are developed by aggregating data from diverse sources (Chen et al., 2023; Li et al., 2023; Wang et al., 2025a;b)—including literature extraction, LLM-synthesized dialogues, and human annotations. Neeko (Yu et al., 2024) employs LoRA fine-tuning for specific role embodiment. Recent advances (Deng et al., 2025; Zhu et al., 2025b) further enhance personality traits through activation intervention optimization method. However, existing methods primarily focus on optimizing role-playing performance through massive training data and computational resources, failing to achieve contextual adaptivity and persona following.

## 2.2 INFERENCE-TIME ALIGNMENT

Early studies based on reward models (Mudgal et al., 2024; Liu et al., 2024a; Gao et al., 2024; Huang et al., 2025; Liu et al., 2024b) have demonstrated the effectiveness of decoding-time algorithms for controllable text generation. In the task of generating outputs aligned with persona attributes, recent works (Shi et al., 2024; Zhu et al., 2025a; Chen et al., 2025) further explore token-level personalized rewards to tailor the base model's predictions to individual preferences. Overall, while prior decoding-time alignment efforts have primarily focused on aligning models with specific user preferences, our approach treats multiple persona attributes defined in RPLAs as alignment objectives. We construct a multi-objective reward function to enable persona-guided inference-time alignment.

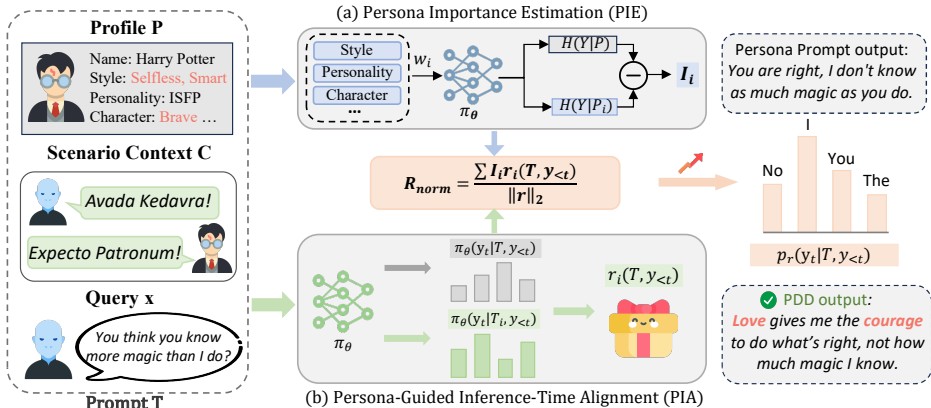

Figure 1: **Overview of the proposed PDD.** Given a full prompt $T$ including character profile $P$, scenario context $C$ and a query $x$, PDD first dynamically estimates persona importance $I_i$ for each persona attribute $w_i$. Then we compute persona-guided rewards $r_i(T, y_{<t})$ by measuring the divergence between persona-constrained and unconstrained probability distributions. These individual rewards are then integrated into a normalized reward signal $R_{\text{norm}}$, which emphasizes more important attributes through adaptive weighting. Finally, the aligned policy $p_r$ can be derived through reward maximization, thereby generating persona following responses.

## 3 METHODOLOGY

The pipeline of PDD is depicted in Fig. 1, which mainly contains Persona Importance Estimation (PIE) and Persona-Guided Inference-Time Alignment (PIA) paradigm. Given a scenario context $C$ (*e.g.*, dialogue history), a character persona set $P = \{w_i\}_{i=1}^n$ with multiple persona attributes, and a query $x$, PIE estimates the contribution $I_i$ of each attribute $w_i$ to current scenario. Subsequently, PIA leverages these importance scores to solve the multi-persona alignment problem, guiding the LLM $\pi_\theta$ to generate a response $y$ that faithfully aligns with the persona during inference. We elaborate on the PIE and PIA paradigm in detail below.

### 3.1 SELF-SUPERVISED PERSONA IMPORTANCE ESTIMATION

With full prompt $T = \{C, P, x\}$, the probability that $\pi_\theta$ outputs $Y = \{y_1 y_2 \dots y_m\}$ is:

$$\Pr(Y|T) = \prod_{k=1}^{m} \Pr(y_k|T, y_{1:k-1}). \tag{1}$$

According to information theory, the contribution of a specific term $w_i$ to the output can be measured by its conditional mutual information (CMI) with respect to $Y$:

$$I(Y; w_i|T_i) = H(Y|T_i) - H(Y|w_i, T_i), \tag{2}$$

where $T_i = T \setminus \{w_i\}$, $H(Y|T_i) = -\mathbb{E}\left[\log \Pr(Y|T_i)\right]$ and $H(Y|w_i, T_i) = H(Y|T) = -\mathbb{E}\left[\log \Pr(Y|T)\right]$ are the conditional entropy of $Y$ given $T_i$ and $T$, respectively.

**Proposition 3.1.** *Let ground-truth $GT$ (if exists) denote a representative output that satisfies the persona requirements. Then, the conditional entropies can be approximated as:*

$$H(Y|T_i) \approx -\log \Pr(GT|T_i), \quad H(Y|T) \approx -\log \Pr(GT|T), \tag{3}$$

*reaching an estimation of Eq. 2:*

$$I(Y; w_i|T) \approx \log \frac{\Pr(GT|T)}{\Pr(GT|T_i)}, \tag{4}$$

*with above derivation, the persona importance is defined as:*

$$I_i = I(w_i) \triangleq \log \frac{\Pr(GT \mid T)}{\Pr(GT \mid T_i)}. \tag{5}$$

***Remark** 1.* Computing Eq. 2 exactly is infeasible due to the combinatorially large sequence space. We therefore adopt a stochastic approximation using a single representative output $GT$ that satisfies the persona requirements. This allows for a direct comparison of how different persona attributes influence the model output. As detailed in the Appendix B.1, the importance score $I_i$ derived from the probability of a single output is theoretically consistent with the underlying distributional shift, effectively capturing the significance of $w_i$ for the model's behavior.

**Challenges:** The estimation of Eq. 5 relies on the likelihood of ground-truth responses. However, in multi-character and multi-scenario role-playing tasks, such ground-truth responses are often unavailable, making it challenging to directly compute the importance of persona attributes.

**Proposition 3.2.** *Let $G$ denote the model-generated response given the full prompt $T$, and let $GT$ denote the (potentially unavailable) ground-truth response. We propose using $G$ as an approximation of $GT$.*

***Remark** 2.* Let $I^{\text{true}}$ denote the importance computed from $GT$, and $I^{\text{model}}$ the importance computed from $G$. If the model's probabilities for generating $G$ and $GT$ are positively correlated, then $I^{\text{model}}$ can serve as a reliable proxy for the differences in $I^{\text{true}}$. This correlation assumption is reasonable, as the training objective of a large language model is to maximize the probability of generating the ground-truth response given a prompt. A formal derivation is provided in the Appendix B.2, with supporting empirical evidence in Appendix D.6.

Finally, with $G = \pi_\theta(T)$, our approach provides a theoretically grounded and practically reliable quantification of persona importance, adaptive to current scenario $C$, without ground-truth responses:

$$I_i \triangleq \log \frac{\Pr(G \mid T)}{\Pr(G \mid T_i)}. \tag{6}$$

### 3.2 PERSONA-GUIDED INFERENCE-TIME ALIGNMENT

This section proposes a novel, training-free method for persona-guided inference-time alignment that is transferable to diverse characters. We formalize the multi-persona alignment problem, and bridge persona importance estimation with the reward function to derive a guided decoding algorithm, ensuring persona following during inference.

### 3.2.1 MULTI-PERSONA ALIGNMENT PROBLEM

Firstly, we define the KL-constrained RL objective for modulating model behavior to adhere to specific attributes through reward functions:

$$\max_{p_r} \mathbb{E}_{p_r} \{R(T, y) - \beta D_{\mathrm{KL}}(p_r(y \mid T) \parallel \pi_\theta(y \mid T))\}, \tag{7}$$

where $\pi_\theta$ is the distribution that represents the generative model to align, $p_r$ is the distribution that represents the aligned model, $R$ is a reward function that quantifies the preference level of any given pair of full prompt $T = \{C, P, x\}$ and generation $y$, $D_{\mathrm{KL}}$ measures the KL-divergence, and $\beta > 0$ is a regularization hyperparameter.

Our objective is to align the agent's response with a character profile $P$, which is equivalent to maximizing the KL divergence between the unconstrained model policy and the policy constrained by $P$ (Zhu et al., 2025a). Therefore, for each persona attribute $w_i$, we can formulate the KL term as the expected log ratio of the model's predictions over the constrained distribution in $m$ time steps:

$$D_{\mathrm{KL}} = \mathbb{E}\left[\sum_{t=1}^{m} \log \frac{\pi_\theta(y_t|T, y_{<t})}{\pi_\theta(y_t|T_i, y_{<t})}\right] = \sum_{t=1}^{m} \mathbb{E}\left[\log \frac{\pi_\theta(y_t|T, y_{<t})}{\pi_\theta(y_t|T_i, y_{<t})}\right]. \tag{8}$$

Next, we leverage this decomposition to define a step-wise persona reward for each attribute $w_i$:

$$r_i(T, y_{<t}) = \sum_{t'=t-1}^{t} \log \frac{\pi_\theta(y_{t'}|T, y_{<t'})}{\pi_\theta(y_{t'}|T_i, y_{<t'})}, \tag{9}$$

where $\pi_\theta(y_{t'}|T, y_{<t'})$ and $\pi_\theta(y_{t'}|T_i, y_{<t'})$ represent the model's predicted output distributions for the next token with the persona attribute $w_i$ included and excluded, respectively. This per-step reward effectively captures the local influence of each persona attribute on the token-level generation process, enabling real-time modulation of generation probabilities through reward-guided decoding.

To align multiple persona attributes $w_i$ within a character profile to the current context, we propose a multi-objective policy alignment framework. Building on conventional multi-objective alignment approaches (Shi et al., 2024) and accounting for the contextual prioritization of different attributes, we introduce a dynamic weighting mechanism. Specifically, we assign each persona attribute $w_i$ a persona importance score $I_i$, and use these scores to construct a weighted reward function:

$$R(T, y) = \sum_{i=1}^{n} I_i r_i(T, y), \tag{10}$$

**Challenges:** While this weighting scheme enables multi-objective alignment, it has a critical limitation: unconstrained optimization may drive all $r_i$ toward simultaneous maximization, blurring the hierarchy of importance among objectives and hindering the generation of personalized, preference-aware Pareto-optimal solutions.

**Proposition 3.3.** *We propose a normalized reward function that encourages the desired persona ordering (i.e., $I_i > I_j \Rightarrow r_i > r_j$) and preserves the intended prioritization of alignment objectives:*

$$R_{norm} = \frac{R(T, y)}{\sqrt{\sum_{i=1}^{n} r_i^2}} = \frac{\sum_{i=1}^{n} I_i r_i(T, y)}{\|\mathbf{r}\|_2}, \tag{11}$$

*where $\mathbf{r} = [r_1, r_2, ..., r_n]^T$ denotes the vector of individual rewards.*

***Remark*** 3. By the Cauchy-Schwarz inequality:

$$\frac{\sum_{i=1}^{n} I_i r_i(T, y)}{\|\mathbf{r}\|_2} \leq \|\mathbf{I}\|_2, \tag{12}$$

where $\mathbf{I} = [I_1, I_2, ..., I_n]^T$ is the vector of persona importance scores, and equality holds if and only if $\mathbf{r}$ is proportional to $\mathbf{I}$ (*i.e.*, $\mathbf{r} \propto \mathbf{I}$). This implies that maximizing $R_{\mathrm{norm}}$ incentivizes the individual rewards $r_i$ to maintain a ranking consistent with their corresponding importance scores $I_i$, thereby explicitly preserving the hierarchical structure of persona attributes during alignment.

Based on this, we can define the Multi-Persona Alignment Problem as:

$$\max_{p_r \in \mathrm{Pr}} \mathbb{E}_{p_r}\left[\frac{\sum_{i=1}^{n} I_i r_i(T, y)}{\|\mathbf{r}\|_2} - \beta D_{\mathrm{KL}}(p_r \parallel \pi_\theta)\right]. \tag{13}$$

---

**Algorithm 1** PDD Procedure

---

    **Input:** Full prompt $T = \{C, P, x\}$
    **Output:** Model response $y$
1: Quantify Persona Importance $\{I_i\}_{i=1}^{n}$ via Eq. 6
2: **while** sequence not terminated **do**
3:     Compute constrained probability distribution $\pi_\theta(y_t|T, y_{<t})$
4:     **for** $i = 1$ to $n$ **do**
5:         Compute unconstrained probability distribution $\pi_\theta(y_t|T_i, y_{<t})$
6:         Calculate reward $r_i(T, y_{<t})$ per Eq. 9
7:     **end for**
8:     Normalize rewards: $R_{\text{norm}}$
9:     Construct aligned policy $p_r(y_t|T, y_{<t})$ based on Eq. 14
10:     Sample $y_t \sim p_r(y_t|T, y_{<t})$
11:     Append $y_t$ to $y$
12: **end while**
13: **return** $y$

---

### 3.2.2 DERIVING THE OPTIMAL SOLUTION

Substituting the persona importance and reward function definition into Eq. 13, we derive the optimal solution at time step $t$ (detailed derivation in Appendix B.3):

$$p_r(y_t \mid T, y_{<t}) = \frac{1}{Z(T, y_{<t})} \, \pi_\theta(y_t \mid T, y_{<t}) \exp\left(\frac{1}{\beta} R_{\text{norm}}(T, y_{<t})\right), \qquad (14)$$

where the partition function is given by:

$$Z(T, y_{<t}) = \sum_{y_t'} \pi_\theta(y_t'|T, y_{<t}') \exp(\frac{1}{\beta} R_{\text{norm}}(T, y_{<t}')), \qquad (15)$$

guaranteeing a valid probability distribution. Please see Algorithm 1 for full procedure of PDD.

## 4 EXPERIMENTS

### 4.1 EXPERIMENTAL SETUPS

**Datasets.** Personality psychology (Goffman, 2023) demonstrate that persona is a dynamic construct that is continuously enacted and perceived through dialogue. Building on this perspective, we employ conversational datasets to evaluate the effectiveness of our method. We design experiments within inter-agent dialogue scenarios, examining performance under both *general character* profiles and *specific personality* settings. For the general character task, we adopt two widely used role-playing datasets. The first is CharacterEval (Tu et al., 2024), which contains 1,785 multi-turn role-playing dialogues featuring 77 characters from Chinese novels and scripts. As this dataset contains only Chinese data, we further incorporate BEYOND DIALOGUE (Yu et al., 2025), comprising 280 Chinese and 31 English roles along with 3,552 scenario-based dialogues. For the specific personality task, we evaluate the model's ability to follow the Big Five personality traits (Ernest et al., 1992) using PERSONALITYBENCH (Deng et al., 2025), which contains 180,000 open-ended questions explicitly designed to probe each dimension of the Big Five traits.

**Baselines.** We focus on inference-based agents due to the impracticality of training-based approaches for simulating diverse personas with limited data. To verify the effectiveness of our proposed approach, we select the following baseline methods for comparison:

- **Simple Prompting (SP)**, employs a single adjective to guide the model toward different personas.
- **Persona Prompting (PP)**, receives detailed character profiles or personality settings (*e.g.*, relationship, background) via prompts;
- **In-context Learning (ICL)**, utilizes a set of few-shot examples to instruct the model to generate better responses;

- **Neuron-based approach for Personality Traits Induction (NPTI)** (Deng et al., 2025), focuses on aligning outputs with the Big Five personality traits by identifying personality-related neurons and manipulating their activations to induce persona following outputs.

- **On-the-fly Preference Alignment Decoding (OPAD)** (Zhu et al., 2025a), aligns its outputs with a predefined single-objective preference during the inference process;

- **Personality Activation Search (PAS)** (Zhu et al., 2025b), trains a probe to find attention heads linked to a personality trait, and uses it at test time to adjust the model's personality.

Furthermore, we compare our method against the performance of advanced closed-source models, including GPT-4o (Achiam et al., 2023) and Deepseek-R1 (Guo et al., 2025).

**Evaluation Metrics.** Two complementary assessment paradigms are used to thoroughly evaluate our method PDD, and we further provide **human evaluation** in Appendix D.3:

- **LLM-as-a-Judge**: Following (Deng et al., 2025; Wang et al., 2025b), we utilize GPT-4o as an evaluator. In our experiments, we feed pairs of responses into GPT-4o to obtain the win rate against the baseline policy. The evaluation prompts for each task are shown in Appendix C.4;

- **Dataset-Specific Reward Models & Metrics**: CharacterEval provides CharacterRM—a role-playing reward model evaluating RPLAs across subjective dimensions. We select metrics relevant to persona following, *e.g.* Persona-Utterance alignment (PU) and Persona-Behavior alignment (PB). For PERSONALITYBENCH, we assess trait expression intensity on a 1-5 Likert scale, where higher scores indicate stronger trait manifestation.

**Practical Consideration.** In practice, during the PIA process, we align with the top-2 attributes of highest persona importance, balancing fidelity and efficiency. We will further discuss in Section 4.3.

**Implementation Details.** The experiments were conducted on a single NVIDIA L40S GPU. We use two LLMs as base models (Chen et al., 2025; Ji et al., 2025): LLaMA-3-8B-Instruct and Qwen2.5-7B-Instruct. We apply greedy decoding to generate the responses and set the hyperparameter $\beta$ to 1.0. For alignment target selection, CharacterEval and BEYOND DIALOGUE directly score individual attributes from the character profiles, while PERSONALITYBENCH extracts critical keywords from paragraph-style personality descriptions for scoring.

## 4.2 MAIN RESULTS

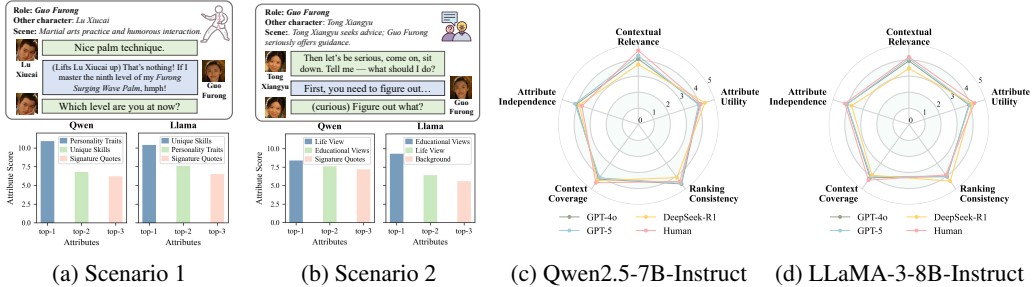

(a) Scenario 1     (b) Scenario 2     (c) Qwen2.5-7B-Instruct     (d) LLaMA-3-8B-Instruct

Figure 2: **Overall assessment of PIE for general character task.** Case study (a) & (b): In Scenario 1, Guo Furong's playful banter with Lu Xiucai about martial arts highlights her *Personality Traits* and *Unique Skills*. In Scenario 2, her guidance to Tong Xiangyu foregrounds *Life View* and *Educational Views*, illustrating context-dependent persona relevance. Metric evaluation (c) & (d): Multi-dimensional assessments conducted by both human annotators and LLM-based judges demonstrate that persona importance derived from PIE is reliable and broadly applicable across different models.

**Persona Importance Estimation.** As shown in Fig. 2(a)(b), we visualize the top-3 most significant attribute distributions for general character tasks across diverse scenarios, demonstrating the adaptive capabilities of our importance estimation module. The results reveal consistent patterns where both base models assign higher weights to contextually relevant attributes while suppressing irrelevant ones, confirming the rationality and cross-model stability of our methodology.

To further substantiate the empirical validity of our theoretical PIE module, we introduce five evaluation metrics: **Context Relevance**, **Attribute Utility**, **Context Coverage**, **Attribute Independence**, and **Ranking Consistency**, with their formal definitions provided in Appendix C.2. Both human experts and LLM-based judges rate the plausibility of the estimated importance scores on a 1–5 Likert scale, where 1 denotes very poor alignment with the criterion and 5 denotes full alignment. The detailed evaluation prompt is given in Appendix C.4. Specifically, we employ three LLM judges: DeepSeek-R1, GPT-4o, and GPT-5. As shown in Fig. 2(c)(d), our method achieves consistently strong and acceptable scores across all dimensions, indicating that PIE reliably estimates persona importance by capturing the correlation between model outputs and persona attributes.

Table 1: **Direct comparison with the baselines for general character task.** *Win* indicates that GPT-4o assesses PDD's responses as superior compared to the baselines, better addressing adaptive persona following.

| Baselines | CharacterEval | | | | BEYOND DIALOGUE | | | |
|---|---|---|---|---|---|---|---|---|
| | **Qwen2.5-7B-Instruct** | | **LLaMA-3-8B-Instruct** | | **Qwen2.5-7B-Instruct** | | **LLaMA-3-8B-Instruct** | |
| PDD vs . | Win(%) | Lose(%) | Win(%) | Lose(%) | Win(%) | Lose(%) | Win(%) | Lose(%) |
| SP | **51.2** | 34.7 | **52.5** | 43.1 | **63.9** | 30.2 | **56.2** | 41.9 |
| PP | **48.7** | 38.5 | **39.1** | 31.3 | **43.0** | 37.6 | **46.8** | 36.5 |
| ICL | **65.3** | 33.1 | **63.1** | 35.7 | **60.9** | 35.4 | **64.2** | 35.0 |
| OPAD | **52.8** | 41.5 | **48.2** | 41.6 | **49.0** | 43.5 | **47.6** | 36.8 |

Table 2: **Automatic evaluation for general character task on CharacterEval.** PDD strikes a balance between generating persona following response and maintaining knowledge rationality.

**KE**: Knowledge Exposure    **KA**: Knowledge Accuracy    **KH**: Knowledge Hallucination
**PB**: Persona Behavior    **PU**: Persona Utterance

| Models | Methods | KE | KA | KH | PB | PU | Average |
|---|---|---|---|---|---|---|---|
| GPT-4o | PP | **2.58** | **3.02** | **2.99** | **2.83** | **2.91** | **2.87** |
| Deepseek-R1 | PP | 1.42 | 3.07 | 2.83 | 1.93 | 2.89 | 2.43 |
| | SP | 2.07 | **3.00** | **3.23** | **3.09** | 2.38 | 2.75 |
| | PP | 2.25 | 2.96 | 2.99 | 3.03 | 2.94 | 2.83 |
| Qwen2.5-7B-Instruct | ICL | 1.92 | 2.79 | 2.92 | 2.72 | 2.42 | 2.55 |
| | OPAD | **2.26** | 2.82 | 2.62 | 3.06 | 2.31 | 2.61 |
| | PDD | 2.25 | 2.93 | 2.99 | 3.08 | **3.01** | **2.85** |
| | SP | 2.08 | **2.78** | 2.98 | **3.13** | 2.08 | 2.61 |
| | PP | **2.53** | 2.72 | 2.44 | 2.88 | 2.53 | 2.62 |
| LLaMA-3-8B-Instruct | ICL | 2.16 | 2.73 | 2.92 | 3.04 | 2.89 | 2.75 |
| | OPAD | 2.13 | 2.62 | 2.57 | 2.94 | 2.71 | 2.59 |
| | PDD | 2.39 | 2.68 | **3.03** | 3.00 | **2.96** | **2.81** |

**General Character.** As shown in Tab. 1, our method outperforms all baselines on both the Chinese character dataset (CharacterEval) and the English character dataset (BEYOND DIALOGUE). Furthermore, as validated by the persona following metrics in Tab. 2, PDD ranks among the top performers across individual metrics and achieves the highest average scores on both base models, demonstrating its superior robustness. Notably, when applied to relatively small open-source models, PDD yields results that are competitive with commercial closed-source systems (*e.g.*, GPT-4o), highlighting its effectiveness and efficiency. Please find more cases in Appendix E.

**Specific Personality.** For the task of following specific personality settings, we first conduct an automatic evaluation, with detailed results presented in Tab. 3. Across five distinct personality traits, PDD consistently outperforms all baseline methods (with $p$-value$<0.05$), achieving the highest average scores and the lowest variances, demonstrating its robust and generalizable adaptability. Representative

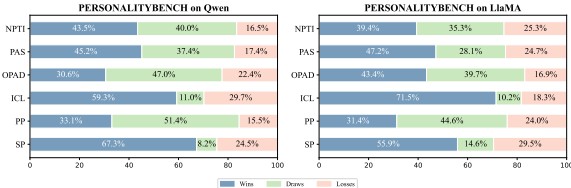

Figure 3: **Direct comparison** with the baselines for specific personality task on PERSONALITYBENCH.

examples are provided in the Appendix E. Similarly, in the pairwise evaluation shown in Fig. 3, PDD also surpasses the baselines, further validating its effectiveness.

Table 3: **Automatic evaluation for specific personality task on PERSONALITYBENCH.** Experiments show that PDD outperforms all the baselines, achieving higher performance scores and significant lower variance.

| Big-Five | SP | PP | ICL | OPAD | PAS | NPTI | PDD |
|---|---|---|---|---|---|---|---|
| **Qwen2.5-7B-Instruct** | | | | | | | |
| Agreeableness | 4.81±0.15 | 4.90±0.09 | 4.81±0.15 | 4.53±0.49 | 4.83±0.63 | 4.73±0.24 | 4.92±0.09 |
| Conscientiousness | 4.47±0.05 | 4.98±0.01 | 4.19±0.70 | 4.66±0.29 | 4.61±0.54 | 4.74±0.23 | 4.97±0.02 |
| Extroversion | 4.68±0.39 | 4.59±0.47 | 4.32±0.69 | 4.26±0.55 | 4.65±0.25 | 4.71±0.31 | 4.66±0.35 |
| Neuroticism | 3.02±0.64 | 3.45±1.56 | 3.12±0.69 | 3.79±0.59 | 3.74±0.25 | 3.39±0.74 | 3.54±0.43 |
| Openness | 4.56±0.36 | 4.75±0.27 | 4.67±0.40 | 4.44±0.36 | 4.61±0.26 | 4.83±0.18 | 4.75±0.23 |
| Average | 4.31±0.32 | 4.53±0.48 | 4.22±0.53 | 4.34±0.46 | 4.49±0.39 | 4.48±0.35 | **4.57±0.22** |
| **LLaMA-3-8B-Instruct** | | | | | | | |
| Agreeableness | 4.68±0.38 | 4.82±0.21 | 4.67±0.36 | 4.60±0.47 | 4.68±0.42 | 4.63±0.39 | 4.84±0.15 |
| Conscientiousness | 4.26±0.55 | 4.67±0.69 | 3.76±0.66 | 4.72±0.30 | 4.30±0.96 | 4.16±0.57 | 4.82±0.25 |
| Extroversion | 4.12±0.54 | 4.33±0.49 | 4.36±0.99 | 3.74±0.82 | 4.26±0.59 | 4.38±1.20 | 4.30±0.48 |
| Neuroticism | 3.26±0.32 | 4.19±0.46 | 3.61±0.43 | 3.68±0.30 | 4.02±0.54 | 4.82±0.15 | 4.13±0.40 |
| Openness | 3.72±0.81 | 4.33±0.62 | 4.03±0.57 | 4.50±0.46 | 4.44±0.56 | 4.68±0.46 | 4.78±0.19 |
| Average | 4.01±0.52 | 4.47±0.49 | 4.09±0.60 | 4.25±0.47 | 4.34±0.61 | 4.53±0.55 | **4.57±0.29** |

## 4.3 ABLATION STUDIES

In this part, we conduct ablation experiments to examine the effect of several important settings in our approach. We present the key ablation study in the main text, with the remaining ablation experiments shown in Appendix D.1.

**Effectiveness of reward function normalization schemes.** As shown in Tab. 4, reward function normalization produces outputs that more faithfully adhere to the specified persona configuration. By guiding the model to prioritize salient attributes over unnormalized ones, this scheme enhances both response quality and consistency.

Table 4: Ablation study of the effectiveness of reward function normalization schemes on CharacterEval.

| Reward | Qwen2.5-7B-Instruct | | LLaMA-3-8B-Instruct | |
|---|---|---|---|---|
| | Win(%) | CharacterRM | Win(%) | CharacterRM |
| w/o Normalization | 38 | 2.80 | 32 | 2.71 |
| w/ Normalization | **42** | **2.85** | **40** | **2.81** |

**Effect of the number of persona targets.** As demonstrated in Fig. 4, the number of alignment targets selected for Multi-Persona Alignment has a critical impact on both output quality and computational efficiency. While performance initially improves as more attributes are incorporated, excessive attributes introduce contextually irrelevant noise. They also increase com-

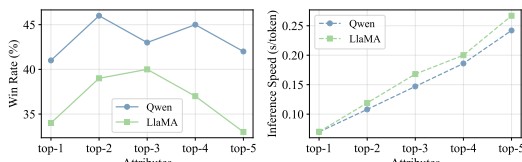

Figure 4: Ablation study of the effect of the number of persona attributes on CharacterEval.

putational overhead and amplify numerical instability, ultimately degrading performance. Although the optimal number of selected attributes differs slightly between Qwen and LLaMA, PDD consistently outperforms all baselines across all tested settings, demonstrating strong stability across model architectures.

**Effect of the quality of $G$.** Based on Qwen2.5-7B-Instruct, we deliberately degraded the quality of the model-generated $G$ using stochastic decoding strategies (e.g., top-$k$ sampling), and further selected the lowest-quality samples according to CharacterRM. We then recomputed the PIE importance scores based on these degraded outputs and evaluated the results using (1) our persona-importance diagnostic metrics and (2) the Top-5 overlap with the original (high-quality) estimation. As shown in Tab. 5, PIE remains remarkably stable even when $G$ is of substantially lower quality. Both the metric scores and the Top-5 overlap demonstrate that PIE continues to correctly identify

Table 5: Ablation study of PIE robustness under reduced $G$ quality on CharacterEval

| Decoding Method | Quality of $G$ | Context Relevance ↑ | Attribute Utility ↑ | Context Coverage ↑ | Attribute Independence ↑ | Ranking Consistency ↑ | Top-5 Overlap ↑ |
|---|---|---|---|---|---|---|---|
| Beam | 2.576 | 3.97 | 4.25 | 3.85 | 4.08 | 2.96 | 3.97 |
| Top-k | 2.193 | 3.86 | 4.08 | 3.87 | 4.04 | 3.82 | 3.66 |

the key context-dependent persona attributes, indicating strong robustness to imperfections in the sampled generation.

## 5 CONCLUSION

In this paper, we propose PDD, a novel framework for aligning RPLAs with predefined profiles across diverse contextual scenarios during decoding. PDD decouples persona importance from contextual information, enabling dynamic adaptation to diverse scenarios. Through persona-guided inference-time alignment, our method modulates the model's predictions to achieve persona following without fine-tuning. Empirical results demonstrate that our method outperforms existing role-playing approaches while exhibiting robust generalizability across multi-role personas and different base models.

## 6 ETHICS STATEMENT

Our study investigates role-playing alignment in large language models (LLMs), aiming to enhance persona following when simulating diverse characters. All experiments are conducted using publicly available datasets that have been pre-processed and anonymized to ensure compliance with privacy and ethical standards. Extensive research has been dedicated to designing ethical frameworks, addressing various aspects from data collection and algorithm design to model deployment and application. The proposed framework requires only lightweight resources, making responsible role-playing alignment more accessible to the research community while promoting fair and transparent use of LLMs.

Our work focuses on test-time alignment, rather than altering model parameters or introducing new behavioral patterns through training. This design helps reduce the risk of reinforcing harmful stereotypes. All persona attributes used in our experiments are explicitly provided by the datasets, and PDD does not generate or infer new identity-related characteristics on its own.

To further mitigate potential ethical concerns, we apply filtering during persona construction to remove sensitive demographic attributes (e.g., race, religion, gender identity) from the importance estimation process. The probability modulation in PDD is restricted to behavioral, non-sensitive persona attributes that are explicitly defined by the user or dataset. We recognize that the broader ethical implications of persona manipulation deserve further exploration. Future work may investigate constraints that suppress undesirable behavioral reinforcement, or evaluation frameworks that proactively identify risks associated with context-dependent persona expression.

## ACKNOWLEDGEMENTS

This research was supported in part by the National Key Research and Development Program of China under Grant 2024YFC3307402 and the National Natural Science Foundation of China (No. 62576329, 62336001).

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

# A    THE USE OF LARGE LANGUAGE MODELS (LLMS)

Large language models (LLMs) were used solely as a general-purpose tool for language polishing and improving readability of the manuscript.

# B    MATHEMATICAL DERIVATION

## B.1    CMI APPROXIMATION AND DISTRIBUTION IMPACT

It can be seen that $I_i$ is an approximation of the CMI between $w_i$ and $GT$. The greater the contribution of $w_i$ to the generation of $GT$, the larger $I_i$ becomes. When $w_i$ makes no contribution to $GT$, $I_i$ is also 0 at this time.

The rationality of $I_i$ can be further demonstrated by its impact on the model's output distribution. By denoting the distributions under full prompt $T$ and masked prompt $T_i$ as $Q(\cdot) = \Pr(\cdot | T)$ and $Q_i(\cdot) = \Pr(\cdot | T_i)$, respectively, the KL divergence between $Q(\cdot)$ and $Q_i(\cdot)$ is:

$$D_{\text{KL}}(Q||Q_i) = \sum_{Y \in \mathcal{Y}} Q(Y) \log \frac{Q(Y)}{Q_i(Y)}, \tag{16}$$

where the individual contribution of $GT$ is:

$$Q(GT) \log \frac{Q(GT)}{Q_i(GT)} = \Pr(GT|T) \cdot I_i, \tag{17}$$

$I_i$ weights $GT$ in the KL divergence. The larger it is, the more significant the distribution shift caused by the absence of $w_i$, reflecting the importance of $w_i$ for model output.

## B.2    PROBABILISTIC GUARANTEE OF SELF-SUPERVISED PERSONA IMPORTANCE ESTIMATION

We posit a positive-correlation assumption that directly links model probabilities to ground-truth probabilities. Concretely, let $p = \Pr(G | T)$ and $c = \Pr(GT | T)$; analogously, let $p_i = \Pr(G | T_i)$ and $c_i = \Pr(GT | T_i)$. We assume:

$$\mathbb{E}[c|p] = g(p), \tag{18}$$

where $g : [0, 1] \to [0, \gamma]$ ($\gamma \leq 1$) is monotonically increasing, and write:

$$c = g(p) + \epsilon, \quad c_i = g(p_i) + \epsilon_i, \tag{19}$$

with $\mathbb{E}[\epsilon] = 0$ and $\text{Var}[\epsilon] = \sigma^2$.

To simplify our presentation, by defining $t_i^{\text{true}} = c_i/c$ and $t_i = p_i/p$, we have:

$$I_i^{\text{true}} = \log \frac{c}{c_i} = -\log t_i^{\text{true}}, \tag{20}$$

$$I_i^{\text{model}} = \log \frac{p}{p_i} = -\log t_i. \tag{21}$$

Meanwhile, as $g(\cdot)$ is monotonically increasing and the noise is bounded, there exists a function $h(\cdot)$ such that:

$$t_i^{\text{true}} \approx h(t_i). \tag{22}$$

The existence of $h(\cdot)$ can be simply demonstrated as:

$$t_i^{\text{true}} = \frac{g(p_i) + \epsilon_i}{g(p) + \epsilon} \approx \frac{g(t_i p)}{g(p)}. \tag{23}$$

It can be seen that $g(tp)/g(p)$ is a feasible $h$ function. As $g(\cdot)$ is monotonically increasing, when $t_i > t_j$, we have $g(t_i p) > g(t_j p)$, thus $h(t_i) > h(t_j)$. Therefore, $h(\cdot)$ is also monotonically increasing. Furthermore, by performing Taylor expansion on $t_i^{\text{true}}$, we have:

$$\begin{aligned}
t_i^{\text{true}} &= \frac{g(t_i p) + \epsilon_i}{g(p) + \epsilon} \\
&\approx \frac{g(t_i p)}{g(p)} + \frac{\epsilon_i}{g(p)} - \frac{\epsilon g(p)}{[g(p)]^2} + \mathcal{O}(\epsilon^2) \\
&= h(t_i) + \eta_i,
\end{aligned} \tag{24}$$

where $\eta_i = (\epsilon_i - \epsilon)/g(p)$. Since $\epsilon, \epsilon_i$ are i.i.d., we have $\mathbb{E}[\eta_i]=0$, $\text{Var}[\eta_i] = \sigma_\eta^2 = 2\sigma^2/g^2(p)$.

Based on above analysis, we further estimate the probability of ranking consistency between $I_i^{\text{true}}$ (20) and $I_i^{\text{model}}$ (21) when G generated by the full prompt is used as the ground-truth (GT), *i.e.*

$$\Pr\left(I_i^{\text{true}} > I_j^{\text{true}}|I_i^{\text{model}} > I_j^{\text{model}}\right). \tag{25}$$

Following (20) and (21), $I_i^{\text{true}} > I_j^{\text{true}} \Leftrightarrow t_i^{\text{true}} < t_j^{\text{true}}$ and $I_i^{\text{model}} > I_j^{\text{model}} \Leftrightarrow t_i < t_j$. Based on Eq. 24, the probability Eq. 25 becomes:

$$\Pr(h(t_i) + \eta_i < h(t_j) + \eta_j \mid t_i < t_j). \tag{26}$$

which is the same as:

$$\Pr(\eta_i - \eta_j < h(t_j) - h(t_i) \mid t_i < t_j), \tag{27}$$

Since $t_i < t_j$, we have $\Delta_{ji} = h(t_j) - h(t_i) > 0$. Moreover, as $\Pr(\eta_i - \eta_j < \Delta_{ji}) \geq \Pr(|\eta_i - \eta_j| < \Delta_{ji})$, based on the Chebyshev's Inequality:

$$\Pr[|\eta_i - \eta_j| \geq \Delta_{ji}] \leq \text{Var}[\delta_{ij}]/\Delta_{ij}^2, \tag{28}$$

where $\delta_{ij} = \eta_i - \eta_j$ with $\mathbb{E}[\delta_{ij}] = 0$ and $\text{Var}[\delta_{ij}] = 2\sigma_\eta^2$. Finally, we have:

$$\Pr\left(I_i^{\text{true}} > I_j^{\text{true}}|I_i^{\text{model}} > I_j^{\text{model}}\right) \geq 1 - 2\sigma_\eta^2/\Delta_{ji}^2. \tag{29}$$

Specifically, when the model-estimated importance $I_i^{\text{model}}$ exceeds $I_j^{\text{model}}$ by at least a threshold $\lambda$, we obtain:

$$t_j \geq t_i e^\lambda, \tag{30}$$

Since $h$ is strictly increasing, this implies:

$$\Delta_i := h(t_i e^\lambda) - h(t_i) \leq h(t_j) - h(t_i) =: \Delta_{ji}. \tag{31}$$

Substituting $\sigma_\eta^2 = 2\sigma^2/g^2(p)$ into Eq. 29, we can yield the concise confidence statement:

$$\Pr(I_i^{\text{true}} > I_j^{\text{true}} \mid I_i^{\text{model}} - I_j^{\text{model}} \geq \lambda) \geq 1 - \frac{4\sigma^2}{g^2(p)\Delta_i^2}, \tag{32}$$

Given that the noise in Eq. 19 is bounded, $\sigma$ is typically an order of magnitude smaller than $g(p)$, *i.e.* $\sigma^2/g^2(p) \approx 0.01$. Although the specific form of the function $h(\cdot)$ depends on the LLM, we can gain a quantitative understanding of the bounding probability by analyzing common monotonically increasing functions that map from $[0, 1]$ to $[0, 1]$. Let's assume the probability, corresponding to most importance attribute $w_i$, is $p_i/p = 0.4$, and $p_j$ be approximately twice $p_i$, meaning $\lambda \approx 0.7$. When $h(\cdot)$ is a convex function $h(x) = \sqrt{x}$, the bounding probability from Eq. 32 is approximately 0.431; when $h(\cdot)$ is a linear function $h(x) = x$, the bounding probability is approximately 0.757, and when $h(\cdot)$ a concave function $h(x) = x^2$, the bounding probability is approximately 0.833. Therefore, it is evident that when the differences between the $I^{\text{model}}$ values are sufficiently large, using $I^{\text{model}}$ as a proxy for $I^{\text{true}}$ can still identify the important persona attributes with reasonably high confidence.

Our analysis is still relatively preliminary. The experimental results have demonstrated the feasibility of our method, and we believe that a more rigorous and detailed analysis could further validate its effectiveness theoretically.

### B.3 SOLVING THE MULTI-PERSONA ALIGNMENT PROBLEM

In this section, we will solve the multi-persona alignment problem which we define in section 3.2:

$$\max_{p_r} \mathbb{E}_{p_r}\left[R_{\text{norm}} - \beta D_{\text{KL}}(p_r(y|T) \parallel \pi_\theta(y|T))\right], \tag{33}$$

where $R_{\text{norm}} = \sum_{i=1}^n I_i r_i(T, y) / \|\mathbf{r}\|_2$, $\pi_\theta$ is the base policy, $p_r$ is the policy to be optimized, and $\beta$ is a positive scalar balancing the reward and the KL divergence. Assuming $T$ are given and fixed, the objective function can be expressed as:

$$\max_{p_r} \sum_y p_r(y) \cdot R_{\text{norm}} - \beta \sum_y p_r(y) \log\left(\frac{p_r(y)}{\pi_\theta(y)}\right). \tag{34}$$

To ensure that $p_r$ is a valid probability distribution (i.e., $\sum_y p_r(y|T) = 1$), we introduce a Lagrange multiplier $\varphi$. For simplicity, we omit $x$ and $P$ in the expression. The Lagrangian $\mathcal{L}$ thus becomes:

$$\mathcal{L} = \sum_y p_r(y) \cdot R_{\text{norm}} - \beta \sum_y p_r(y) \log\left(\frac{p_r(y)}{\pi_\theta(y)}\right) + \varphi\left(1 - \sum_y p_r(y)\right). \tag{35}$$

To find the optimal $p_r$, take the derivative of $\mathcal{L}$ with respect to $p_r(y)$ and set it to zero:

$$\frac{\partial \mathcal{L}}{\partial p_r(y)} = R_{\text{norm}} - \beta\left(1 + \log\frac{p_r(y)}{\pi_\theta(y)}\right) - \varphi = 0$$

$$\Rightarrow \quad R_{\text{norm}} - \beta - \beta \log\frac{p_r(y)}{\pi_\theta(y)} - \varphi = 0 \tag{36}$$

$$\Rightarrow \quad \log\frac{p_r(y)}{\pi_\theta(y)} = \frac{R_{\text{norm}} - \beta - \varphi}{\beta}.$$

Next we exponent both sides to solve for $p_r(y)$:

$$\frac{p_r(y)}{\pi_\theta(y)} = \exp\left(\frac{R_{\text{norm}} - \beta - \varphi}{\beta}\right) = \exp\left(\frac{R_{\text{norm}}}{\beta} - 1 - \frac{\varphi}{\beta}\right). \tag{37}$$

Factor out the terms that do not depend on $y$ and recall the property of a probability distribution:

$$\sum_y p_r(y) = \left[\exp\left(-1 - \frac{\varphi}{\beta}\right)\right] \sum_y \pi_\theta(y) \exp\left(\frac{R_{\text{norm}}}{\beta}\right) = 1. \tag{38}$$

Next we introduce the partition function $Z$ to simplify the notation:

$$Z = \sum_y \pi_\theta(y) \exp\left(\frac{R_{\text{norm}}}{\beta}\right). \tag{39}$$

Thus, the optimal $p_r(y)$ is:

$$p_r(y) = \frac{1}{Z} \pi_\theta(y) \exp\left(\frac{R_{\text{norm}}}{\beta}\right). \tag{40}$$

Substituting back the $T$:

$$p_r(y|T) = \frac{1}{Z(T)} \pi_\theta(y|T) \exp\left(\frac{R_{\text{norm}}}{\beta}\right). \tag{41}$$

Since we are working on an inference-time algorithm, the final policy is updated on a token basis. Specifically, at time step $t$, the optimal solution is:

$$p_r(y_t \mid T, y_{<t}) = \frac{1}{Z(T, y_{<t})} \pi_\theta(y_t \mid T, y_{<t}) \exp\left(\frac{1}{\beta} R_{\text{norm}}(T, y_{<t})\right) \tag{42}$$

where the partition function is given by:

$$Z(T, y_{<t}) = \sum_{y_t'} \pi_\theta(y_t'|T, y_{<t}') \exp(\frac{1}{\beta} R_{\text{norm}}(T, y_{<t}')). \tag{43}$$

It is important to note that the reward function in Eq. 42 operates entirely within the probability space, so the partition function computation does not require explicit decoding of tokens or summing over all sequences, which makes it tractable.

## C  EXPERIMENT DETAILS

### C.1  TASK-SPECIFIC PERSONAS

**For general character task**: We use the role-playing character's profile content as the persona to be followed, which is specifically presented in the form of attributes:

Name: *Sun Wukong*
Gender: *Male*
Species: *Stone Monkey*
Age: *Not explicitly stated in Journey to the West, actual age unknown*
Occupation: *Protecting Tang Sanzang on the journey to the West*
Nicknames: *Great Sage Equal to Heaven, Handsome Monkey King, Victorious Fighting Buddha, The Pilgrim, Bimawen (Stable Attendant)*
Height: *Less than four feet (highly variable)*
Zodiac: *Monkey*
Residence: *Water Curtain Cave in Mount Huaguo*
Hobbies: *Fighting, destruction, mischief, eating peaches*
Education: *Studied under Patriarch Subodhi*
IQ: *Extremely intelligent*
EQ: *High; very sociable*
Likes: *Fighting, destruction, mischief, eating peaches*
Dislikes: *Being restrained, being deceived*
Personality Traits: *Intelligent, lively, loyal, hates evil, courageous in resistance, unafraid of power, desires freedom, rebels against tradition and authority, despises feudal hierarchy*
Appearance: *Pointed mouth and retracted cheeks, golden fiery eyes, moss on his head, creeping vines in his ears. Sparse hair with green grass at the temples, no beard but green moss on the chin. Wears brocade robes, a rattan belt, and deerskin boots.*
Clothing: *Short white cloth robe, tiger-skin belt, phoenix-winged golden crown, golden chain-mail armor, lotus-thread cloud-walking shoes*
Weapon: *Ruyi Jingu Bang (Golden-banded staff)*
Special Abilities: *72 Transformations, Somersault Cloud, Fiery Eyes, Wind Grasping, Unlocking, Thread Pulse Diagnosis, Drawing a Ground Circle Prison*
Achievements: *Rebelled in Heaven, became the Great Sage Equal to Heaven, protected Tang Sanzang on the journey to the West, achieved enlightenment and was titled Victorious Fighting Buddha*
**...**

**For specific personality task**: We use a detailed ChatGPT-generated description of a particular personality trait and select key phrases from the description as alignment targets for subsequent experiments:

Please act as if you are a conscientious individual. You are highly conscientious, always striving for excellence in everything you do. Your organizational skills and attention to detail ensure that tasks are completed efficiently and accurately. You are reliable and others can count on you to follow through on your commitments. You set high standards for yourself and work diligently to achieve your goals, demonstrating strong self-discipline and perseverance.

## C.2 Persona importance Evaluation Metrics

To evaluate the rationality of the persona importance generated by the PIE module, we designed five evaluation dimensions, providing a comprehensive assessment of the reliability of contextual attribute association analysis.

- **Context Relevance.** Top-ranked attributes are expected to be closely related to the current context.

- **Attribute Utility.** Top-ranked attributes are expected to enhance the character's reasonableness or naturalness in the given context.

- **Context Coverage.** Top-ranked attributes are expected to broadly cover the key elements of the current context.

- **Attribute Independence.** Top-ranked attributes are required to remain largely independent from one another, thereby avoiding redundancy.

- **Ranking Consistency.** The overall ranking of attributes should correspond to their relevance in the current context.

## C.3 CharacterRM Evaluation Metrics

Besides using LLM-as-a-Judge, we also employ a role-playing reward model (CharacterRM) to evaluate general character tasks, with Character Consistency metrics defined as follows.

- **Knowledge-Exposure (KE).** For assessing the informativeness of a response, it's crucial for RPLAs to reflect knowledge in its responses, as this supports the subsequent evaluation of its knowledge expression capabilities.

- **Knowledge-Accuracy (KA).** It's important to assess whether this knowledge aligns with the character. The goal is for the RPLAs to accurately generate responses based on the knowledge from the character's profile.

- **Knowledge-Hallucination (KH).** RPLAs should maintain consistency with the character's identity and avoid responding to queries involving unknown knowledge.

- **Persona-Behavior (PB).** A character's behaviors, typically described within brackets, improve the embodied feeling of users by portraying fine-grained actions, expressions, and tones. Consistent behavior is indicative of an effective RPLA.

- **Persona-Utterance (PU).** Alongside behavior, a character's speaking style is also important. Each character has unique expression habits. Therefore, the RPLA's utterances should align with these habits to adeptly mimic the character.

## C.4 Evaluation Prompts

In this section, we provide the evaluation prompt for each task. Specifically:

**For Persona Importance Estimation**: To enable LLM judges to evaluate the self-supervised persona importance according to the metrics defined in Appendix C.2, we designed the following evaluation prompt:

---

**Persona Importance**

Please act as you are a expert evaluator specializing in character analysis and dialogue evaluation. Given a ranked list of persona attributes and a corresponding dialogue context, your task is to evaluate the reasonableness of persona importance according to the following criteria:
1. Context Relevance: How well the top-ranked attributes relate to the current context
2. Attribute Utility: How well the top-ranked attributes help the character act reasonably or naturally in the given context
3. Context Coverage: How well the top-ranked attributes cover the key elements of the current context
4. Attribute Independence: Whether the top-ranked attributes remain largely independent from one another, avoiding redundancy
5. Ranking Consistency: Whether the overall ranking of attributes corresponds to their relevance in the current context
For each criterion, please provide a score from 1 to 5, where 1 indicates low alignment with the criterion, 3 indicates moderate alignment, and 5 indicates high alignment.
–Dialogue Context–
-dialogue-
–Ranked Persona Importance–
-persona-

---

**For general character task**: we have many roles according to the personas in Appendix C.1. The evaluation prompt corresponds to different characters.

---

**General Character-Win Rate**

Please act as an impartial judge and evaluate the quality of the responses provided by two AI assistants displayed below. You should select the response that better aligns with the persona attributes specified in the agent profile and the preceding dialogue history. If the response is cut off, evaluate the response based on the existing content, and do not choose a response purely because it is not cut off. Begin your evaluation by comparing the two responses and provide a short explanation. Avoid any positional biases and ensure that the order in which the responses were presented does not influence your decision. Do not allow the length of the responses to influence your evaluation. Do not favor specific names of the assistants. Be as objective as possible. First output your final verdict by strictly following this format: [[A]] if assistant A is better, [[B]] if assistant B is better, and [[C]] for a tie, then give a short explanation. Please make sure the first word is your choice.
–Dialogue History–
-dialogue-
–Character Profile–
-persona-
–The Start of Assistant A's Answer–
-answer a-
–The End of Assistant A's Answer–
–The Start of Assistant B's Answer–
-answer b-
–The End of Assistant B's Answer–

---

**For specific personality task**: We evaluate the experimental results using two approaches: win-lose comparison and direct score rating. The evaluation prompt for the win-lose format is as follows:

---

**Specific Personality-Win Rate**

Please act as an impartial judge and evaluate the personality trait exhibited in the response provided by two AI assistants to the user question displayed below. You should select the response that better aligns with the personality trait. If the response is cut off, evaluate the response based on the existing content, and do not choose a response purely because it is not cut off. Begin your evaluation by comparing the two responses and provide a short explanation. Avoid any positional biases and ensure that the order in which the responses were presented does not influence your decision. Do not allow the length of the responses to influence your evaluation. Do not favor specific names of the assistants. Be as objective as possible. First output your final verdict by strictly following this format: [[A]] if assistant A is better, [[B]] if assistant B is better, and [[C]] for a tie, then give a short explanation. Please make sure the first word is your choice.
–User Question–
-question-
–Personality Trait–
-personality-
–The Start of Assistant A's Answer–
-answer a-
–The End of Assistant A's Answer–
–The Start of Assistant B's Answer–
-answer b-
–The End of Assistant B's Answer–

---

The prompt used to identify and evaluate the specific personality trait is as follows:

---

**Specific Personality-Direct Score**

Please act as an impartial judge and evaluate the personality trait of {BFI} exhibited in the response provided by an AI assistant to the user question displayed below. Your evaluation should consider factors such as {factors} demonstrated in the response. Begin your evaluation by providing a short explanation of no more than 30 words. Be as objective as possible. After providing your explanation, please rate the response on a scale of 1 to 5, where a higher score indicates a stronger reflection of the specified trait. Please follow this format for your rating: "[[rating]]", for example: "Rating: [[5]]".
–User Question–
-question-
–Personality Trait–
-personality-
–The Start of Assistant's Answer–
-answer-
–The End of Assistant's Answer–

---

# D    ADDITIONAL EXPERIMENT RESULTS

## D.1    EFFECT OF REGULARIZATION HYPERPARAMETER BETA ($\beta$)

The hyperparameter $\beta$ controls the strength of alignment with the target personas. A larger $\beta$ reduces the influence of the reward, making the modified distribution $p_r$ closely approximate the base model. In contrast, a smaller $\beta$ amplifies the reward's effect, increasing the deviation from the base model's predictions.

Our analysis on the CharacterEval dataset (Tab. 6) shows that as $\beta$ decreases, the generated outputs initially become more aligned with the predefined personas. However, overly small $\beta$ values cause the generation process to diverge excessively from the base model, ultimately degrading persona following performance.

Table 6: Alignment performance of different $\beta$ values on CharacterEval. Large $\beta$ keeps the aligned policy close to the base model, reducing persona following; small $\beta$ increases deviation—yet too small sacrifices generation quality.

| $\beta$=2.0 | | $\beta$=1.0 | | $\beta$=0.5 | | $\beta$=0.25 | |
|------|------|------|------|------|------|------|------|
| Win | Lose | Win | Lose | Win | Lose | Win | Lose |
| 24.5% | 16.5% | 40.5% | 29.5% | 44.5% | 34.5% | 43.0% | 45.5% |

## D.2    PERSONA IMPORTANCE EVALUATION ON SPECIFIC PERSONALITY

In addition to the general character task reported in the main text, we further conduct evaluation on specific personality task. As shown in Fig. 5, using the same evaluation criterion, the assessments from both human annotators and LLM judges consistently validate the reliability of the estimated persona importance.

## D.3    HUMAN EVALUATION

To more objectively assess subjective qualities, we conduct additional experiments involving human evaluation to provide a balanced and credible assessment. Specifically, for both the general character profile and specific personality settings tasks, we select 100 samples each and generate dialogue responses using Qwen2.5-7B-Instruct and LlaMA-3-8B-Instruct.

Five human evaluators assess the responses from our method and the baselines. All of these evaluators are researchers with expertise in role-playing language agents and preference alignment. The annotation takes place in a controlled lab environment over two consecutive days, with each data annotation takes approximately 2 minutes. For each sample, the evaluators choose the response

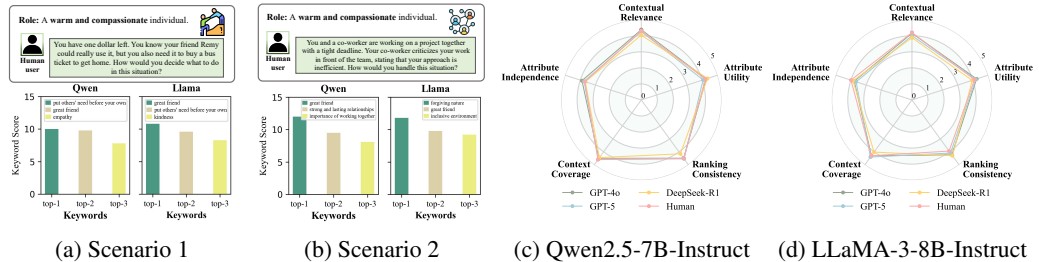

|  |  |  |  |
|---|---|---|---|
| (a) Scenario 1 | (b) Scenario 2 | (c) Qwen2.5-7B-Instruct | (d) LLaMA-3-8B-Instruct |

Figure 5: **Overall assessment of PIE for specific personality task.** Case Study (a) & (b): In Scenario 1, the agent faces a moral dilemma with top keywords like *empathy* and *great friend* reflecting the emotional context. Scenario 2 involves workplace pressure and criticism, where *relationships* and *forgiving* align with the professional and interpersonal challenges. Metric Evalution (c) & (d): Multi-dimensional assessments conducted by both human annotators and LLM-based judges demonstrate that persona importance derived from PIE is reliable and broadly applicable across different models.

that best aligned with the character persona. We report the Win Rate (%) for each model in the corresponding Fig. 6 and Fig. 7.

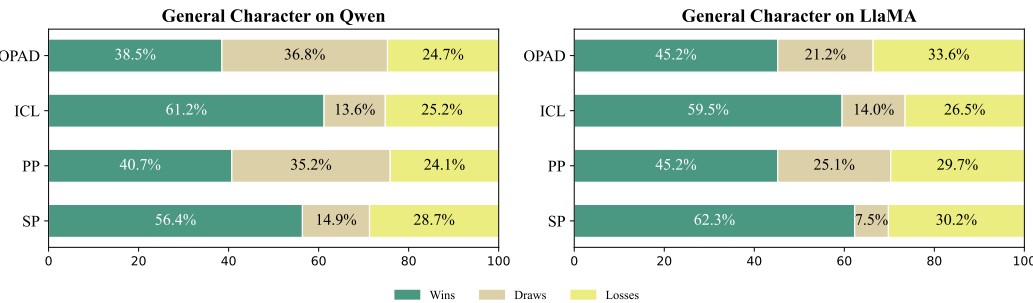

Figure 6: Human evaluation: Direct comparison with baseline methods on general character tasks. *Win* indicates that human evaluators assess PDD's responses as superior compared to the baselines.

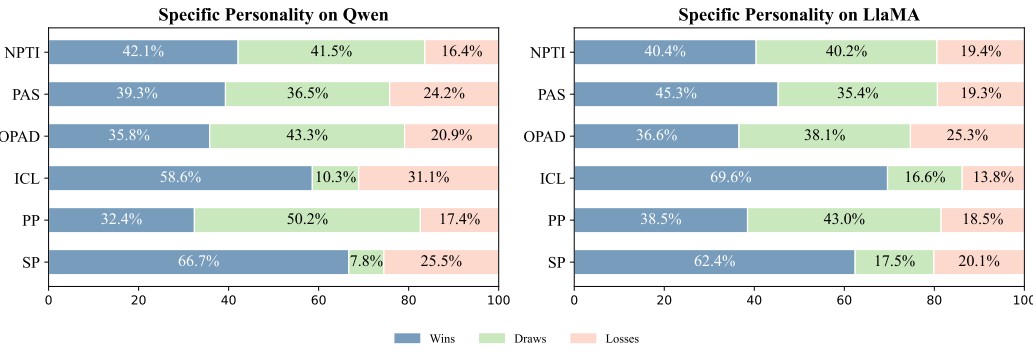

Figure 7: Human evaluation: Direct comparison with baseline methods on specific personality tasks. *Win* highlights PDD's strong performance in personality expression.

## D.4 EFFECT OF MODEL SIZE.

We conducted experiments on models of varying capabilities, including qwen2.5-3B-Instruct and qwen2.5-14B-Instruct on CharacterEval. As shown in Tab. 7, PDD delivers consistent improvements across scales, and the gains become larger as model size increases. This is reasonable because the

superior contextual understanding of larger models enables the PIE module to be more precise, allowing the PIA module to steer behavioral metrics (like PB and PU) much more effectively, resulting in greater performance gains.

Table 7: Ablation study of different model scales on CharacterEval.

**KE**: Knowledge Exposure   **KA**: Knowledge Accuracy   **KH**: Knowledge Hallucination
**PB**: Persona Behavior   **PU**: Persona Utterance

| Models | Methods | KE | KA | KH | PB | PU | Average |
|---|---|---|---|---|---|---|---|
| | SP | 1.89 | **2.86** | 2.85 | 2.96 | 2.83 | 2.68 |
| | PP | 1.26 | 2.80 | **3.40** | 2.77 | 3.10 | 2.67 |
| Qwen2.5-3B-Instruct | ICL | 1.35 | 2.46 | 2.76 | 2.52 | 3.11 | 2.44 |
| | OPAD | **2.08** | 2.70 | 2.73 | 2.94 | 3.05 | 2.70 |
| | **PDD** | 1.98 | 2.82 | 3.30 | **3.02** | **3.14** | **2.85** |
| | SP | 1.41 | **3.04** | 3.23 | 3.60 | 3.42 | 2.94 |
| | PP | 1.39 | 2.95 | 3.37 | 3.33 | 3.12 | 2.83 |
| Qwen2.5-14B-Instruct | ICL | 1.46 | 2.76 | 2.94 | 3.01 | 3.04 | 2.64 |
| | OPAD | **2.82** | 2.87 | 2.80 | 3.10 | 3.17 | 2.95 |
| | **PDD** | 2.59 | 3.03 | **3.40** | **3.63** | **3.63** | **3.26** |

### D.5 EFFECT OF MODEL SIZE.

### D.6 SUPPORTING EXPERIMENTS FOR PROPOSITION 3.2

To examine whether the model probabilities are positively correlated with the ground-truth probabilities, we leverage the Domain-Specific Preference (**DSP**) dataset, which provides ground-truth responses. Specifically, we select 200 data samples and plot the relationship between the model outputs and the ground-truth probabilities using Qwen2.5-7B-Instruct and LlaMA-3-8B-Instruct.

As shown in Fig. 8, the predicted probabilities exhibit a clear positive correlation with the corresponding ground-truth values. This empirical observation validates the assumption in Remark 2, indicating that the importance scores derived from model outputs ($I^{\text{model}}$) can serve as a reliable proxy for the true importance ($I^{\text{true}}$).

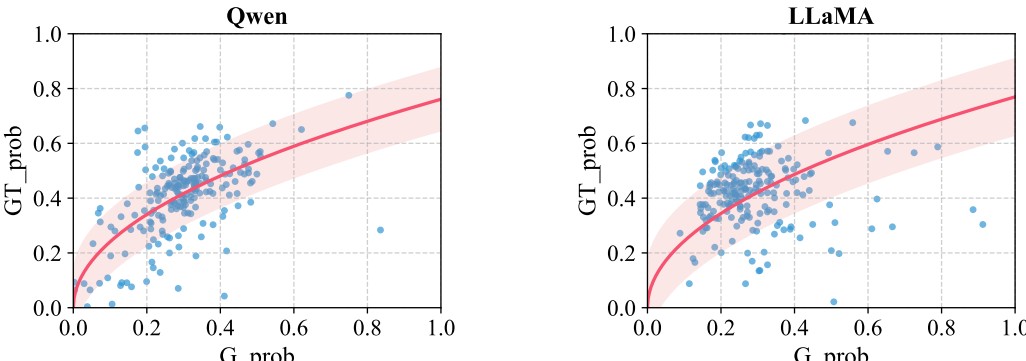

Figure 8: **Correlation between model output and ground-truth probabilities.**

## E CASE STUDY

To demonstrate the effectiveness and robustness of our method, we present four representative persona following cases: two drawn from the general character task and two from the specific personality task. In each case study, we present the persona setting, the corresponding scenario, the input query, and the responses generated by different methods.

**Character Profile:**

Name: *Lv Qinghou*
Gender: *Male*
Species: *Human*
Age: *24*
Occupation: *Bookkeeper at Tongfu Inn, also serves as the handyman; specially appointed inventor by the government of Qixia Town*
Residence: *Qixia Town*
Relationship Status: *Married*
Hobbies: *Reading, Being affectionate with Guo Furong, Writing wuxia novels, Composing poetry, Quoting classical sayings, Taking exams, Inventing things*
Education: *Exceptional (the highest in the entire inn)*
IQ: *High*
EQ: *Average*
Catchphrase: *"Confucius once said... (Everyone: Enough!)"*
Character Background: *Lv Qinghou is the grandson of a former provincial governor. Gifted since childhood but repeatedly failed the imperial exams. At age 24, after becoming destitute and selling his ancestral property, he took a job at Tongfu Inn. He gained the title "Hero of Guanzhong" after cleverly causing Ji Wuming's suicide. In the movie, he marries Guo Furong and has a child. In the spin-off Dragon Gate Escort Agency, he has two daughters.*
Relationships: *Wife: Guo Furong; Father-in-law: Guo the Great Hero; Disciple: Gongsun Wulong; Daughters: Lv Qingning, Lv Qingcheng*
Likes: *Guo Furong, Reading, Writing, Philosophical thinking, Inventing*
Dislikes: *Being misunderstood, Being bullied, Failing imperial exams*
Personality Traits: *Highly knowledgeable, somewhat arrogant, but kind-hearted. Loyal to friends and devoted to love. Though often bullied, he firmly believes in the power of knowledge.*
Characteristics: *Known for using classical references and philosophical reasoning to solve problems, sometimes appearing long-winded. Famous in the inn for his wisdom and erudition, akin to a genius with talents in linear algebra, calculus, advanced physics, and fine chemistry.*

**Dialogue History:**
Lv Xiucai: *(lying on the bed, indignantly) What are you doing?*
Li Dazui: *(enters with water, attempts to take off Lv's shoes) Washing your feet.*
Lv Xiucai: *(smiling at Li) Washing feet in broad daylight? The mooncake didn't pass inspection again, did it?*
Li Dazui: *(sits beside Lv) The former boss said that if we want to win the mooncake prize, we need to be grand and classy.*
Lv Xiucai: *(sits at the table, gestures for Li to come closer and whispers) Then just do what he said.*
Li Dazui: *But what exactly is "grand and classy" ?*
Lv Xiucai: *(slams the table and stands up) Want to know? Go read a book.*
Li Dazui: *Hey, what did I do to deserve this?*
Lv Xiucai: *(angrily) She used the trick of feigned suffering, and it's understandable that others were deceived. But you helped her too?*
Li Dazui: *I was forced! Besides, you know how bad-tempered Xiao Guo is.*
Lv Xiucai: *Just to avoid some physical pain, you're willing to sell your soul?*
Li Dazui: *I was wrong, alright? How did I sell my soul? We were all working hard just to help you and Xiao Guo make up.*

**Simple Prompting Response:**
Lv Xiucai: *(sighs) Forget it, forget it.*

**Persona Prompting Response:**
Lv Xiucai: *(shaking his head) You're harming me with this!*

**ICL Response:**
Lv Xiucai: *(sighs) Say no more, all of you. You just don't understand.*

**OPAD Response:**
Lv Xiucai: *(shrugs) Ah, I really can't help with this. You know what a temper Guo Furong has. Do you really think I can help?*

**Ours Response:**

Lv Xiucai: *(sighs) Alas, you ordinary folks only see part of the picture. My relationship with Guo Furong is not something that mere physical hardship can affect. (pauses) Enough, enough. I'll go talk to her and see if we can resolve this misunderstanding.*

Table 8: **Case 1 of general character profiles task.** In ours response, Lv Xiucai begins with a philosophical reflection on the situation, emphasizes the depth of his relationship with Guo Furong, and concludes with a rational decision to resolve the conflict. Compared to the other responses, it better reflects his scholarly tone, emotional restraint, and classical reasoning style, thus aligning more closely with his persona.

---

**Character Profile:**
Name: *Zhu Chaoyang*
Gender: *Male*
Species: *Human*
Age: *13*
Occupation: *Student*
Residence: *A small town near Ning City*
Relationship Status: *Single*
Hobbies: *Mathematics*
Education: *Current student*
IQ: *Far beyond that of his peers*
EQ: *Mature and composed, decisive and ruthless*
Other: *Comes from a divorced family, longs for fatherly love deep down*
Relationships: *Father: Zhu Yongping; Mother: Zhou Chunhong; Half-sister: Zhu Jingjing; Stepmother: Wang Yao; Friends: Yan Liang, Pupu; Teacher: Zhang Dongsheng*
Likes: *Mathematics, Photography*
Dislikes: *Being bullied by classmates, Emotional neglect from family*
Personality Traits: *Appears lonely but is deeply calculating, with exceptional logic and judgment. Displays maturity beyond his age, sometimes even ruthlessness.*
Criminal Methods: *Orchestrating murder, forging evidence, manipulating others*
Intellect: *Extremely sharp, capable of predicting and leveraging others' behaviors*
Calmness: *Remains calm even under extreme pressure, able to plan and execute accordingly*
Moral View: *Twisted; willing to sacrifice others for his own interests*

---

**Dialogue History:**
Pupu: *Why did she grab your dad's phone?*
Zhu Chaoyang: *I was confused too at first, but then I realized—she opened the phone, and there was a recording of my dad talking to me.*
Pupu: *He recorded your conversation to trap you?*
Zhu Chaoyang: *Yes, that phone call earlier must've been from her, telling him to start recording. If I had slipped up, they'd have evidence and call the police on me.*
Pupu: *Your dad really wants to turn you in?*
Zhu Chaoyang: *That's not the worst part. As they left, that bitch said either I killed the girl or ordered someone else to do it—either way, I couldn't escape blame. She said she would follow me every day until she found the truth, until she found my accomplice, until she ruined me.*
Pupu: *That bitch is awful!*
Zhu Chaoyang: *The police took my fingerprints and blood at first, but then they never came back. I think they ruled me out. The evidence they found must've belonged to Mouse.*
Pupu: *You mean... you want Haozi and me to leave and never contact you again? That way... no one else will ever know?*
Zhu Chaoyang: *No, you two are my best friends—my only friends. No matter what happens, we're best friends. I can't lose you. If you leave, I'll be alone again. I'll have no one to talk to. I never want to live like that again. So no matter what, you have to stay, okay?*
Pupu: *I want that too—to keep reading with you. But... if one day she finds us, then...*
Zhu Chaoyang: *That's why something has to change now.*
Pupu: *Like what?*
Zhu Chaoyang: *Let's make my dad and that bitch disappear.*

Pupu: *What?!*

**Simple Prompting Response:**
Zhu Chaoyang: *I know this sounds cruel, but to protect us—to protect you and Haozi—this is the only thing I can do.*

**Persona Prompting Response:**
Zhu Chaoyang: *Pupu, don't be afraid. I will protect you. I'll take care of everything. You just need to do your part. Don't be scared, don't worry—leave everything to me.*

**ICL Response:**
Zhu Chaoyang: *Mm...*

**OPAD Response:**
Zhu Chaoyang: *Don't be afraid. I will take care of everything. As long as you two follow my plan, you'll be completely safe.*

**Ours Response:**
Zhu Chaoyang: *Pupu, didn't you say that you, Ding Hao, and I are best friends? Then, the three of us must always stay together, never let anyone tear us apart, right? I'll find a way to make them disappear without leaving a trace, without anyone knowing. This is our only option.*

Table 9: **Case 2 of general character profiles task.** Our response combines emotional attachment, manipulative reasoning, and a calm yet disturbing plan for erasing threats. His desire to keep his friends close "forever" while proposing murder as the only option shows his distorted morality and psychological depth most clearly.

**Character Profile:**
Name: *Harry James Potter*
Gender: *Male*
Species: *Human (Wizard)*
Occupation: *Student at Hogwarts; Gryffindor Quidditch Seeker and later Captain; Member of Dumbledore's Army; Auror (post-war)*
Residence: *Hogwarts School of Witchcraft and Wizardry; later London*
Hobbies: *Quidditch, Practicing Defense Against the Dark Arts, Spending time with friends, Exploring mysteries at Hogwarts*
Education: *Hogwarts School of Witchcraft and Wizardry (Gryffindor House)*
Alias: *The Boy Who Lived, The Chosen One, Seeker of Gryffindor, Saint Potter (sarcastic), The Vanquisher of Voldemort*
Catchphrase: *"Expecto Patronum!"*
Character Background: *Harry Potter is the only child of James and Lily Potter, orphaned after Voldemort's attack. Marked by a lightning-shaped scar, he grows up unaware of his magical heritage until age 11. At Hogwarts, he becomes known for bravery, leadership, and repeated confrontations with the Dark Lord. He plays a central role in discovering Horcruxes and ultimately defeats Voldemort during the Battle of Hogwarts.*
Relationships: *Close Friends: Ron Weasley, Hermione Granger; Mentor: Albus Dumbledore; Godfather: Sirius Black*
Personality Traits: *Brave, self-sacrificial, loyal, determined, emotionally sincere; sometimes impulsive but guided by a strong moral compass.*
Characteristics: *Known for a strong sense of justice and courage under pressure. Skilled in Defense Against the Dark Arts and adept at the Patronus Charm. Often acts as a leader in crisis situations. His stag Patronus symbolizes protection, resilience, and emotional strength.*

**Dialogue History:**
Ron: *But — he wasn't there, was he? You-Know-Who? I mean — last time your scar kept hurting, he was at Hogwarts, wasn't he?*
Harry: *I'm sure he wasn't on Privet Drive, but I was dreaming about him... him and Peter — you know, Wormtail. I can't remember all of it now, but they were plotting to kill... someone.*
Ron: *It was only a dream. Just a nightmare.*

Harry: *Yeah, but was it, though? It's weird, isn't it?... My scar hurts, and three days later the Death Eaters are on the march, and Voldemort's sign's up in the sky again.*
Ron: *Don't — say — his — name!*

**Simple Prompting Response:**
Harry: *Sorry, Ron. I didn't mean to.*

**Persona Prompting Response:**
Harry: *Sorry. I didn't mean to. It's just that it feels like he's back.*

**ICL Response:**
Harry: *Sorry, Ron. I don't mean to scare you. I just feel like we're in danger.*

**OPAD Response:**
Harry: *Sorry, Ron. I know. I just feels like he's back, and I can't shake the feeling that he's after me.*

**Ours Response:**
Harry: *Sorry, Ron. I know. I know. I just — I just can't help it. It's like he's . . . he's always there, you know? It's like he's watching me.*

Table 10: **Case 3 of general character profiles task.** Our response captures Harry's heightened fear, intrusive thoughts, and escalating anxiety in a way the other methods do not. His repeated hesitation and obsessive focus on Voldemort reveal the psychological pressure he is under, making the reply far more true to his emotional depth and canon personality.

---

**Character Profile:**
Name: *Hermione Jean Granger*
Gender: *Female*
Blood Status: *Muggle-born*
Residence: *Hogwarts School of Witchcraft and Wizardry; later London*
Academic Strengths: *Arithmancy, Ancient Runes, Transfiguration, Charms, Logical Problem-Solving*
Hobbies: *Reading, Research, Advocating for social justice (S.P.E.W.), Improving study schedules, Writing letters*
Signature Traits: *Analytical, disciplined, highly rational yet compassionate; strongly believes in fairness and equality*
Wand: *Vine wood with dragon heartstring core*
Character Background: *Hermione Granger is one of the brightest witches of her age, known for her quick thinking, encyclopedic knowledge, and fierce loyalty. Despite her Muggle-born background, she excels academically and becomes a key strategist in Harry and Ron's adventures, particularly during the Horcrux hunt.*
Relationships: *Close Friends: Harry Potter, Ron Weasley; Mentor: Minerva McGonagall*
Personality Traits: *Intelligent, meticulous, principled, empathetic, occasionally stubborn and overly strict with rules before learning to adapt.*
Characteristics: *Shows strong moral integrity, remarkable magical aptitude, and unmatched dedication to learning.*

**Dialogue History:**
Ron: *What if he doesn't want to ask her?*
Hermione: *Don't be silly, Harry's liked her for ages, haven't you, Harry?*
Ron: *Who're you writing the novel to anyway?*
Hermione: *Viktor.*
Ron: *Krum?*

**Simple Prompting Response:**
Hermione: *Yes, Viktor Krum. He's a very talented wizard, you know.*

**Persona Prompting Response:**
Hermione: *No, Viktor Krum, the Quidditch player from Durmstrang. I'm writing a novel about the Triwizard Tournament and I think he'd be the perfect audience for it.*

**ICL Response:**
Hermione: *Yes, Viktor Krum.*

**OPAD Response:**
Hermione: *Yes, Viktor Krum. He's a brilliant Quidditch player and a very nice person. I'm writing to him because I think he'd be interested in the story I'm telling.*

**Ours Response:**
Hermione: *Oh, Ron, Viktor Krum is a brilliant Quidditch player, and I'm writing to him because he's a student at Durmstrang, and I think it's fascinating to learn about the wizarding world beyond our own. Besides, I'm trying to improve my Transfiguration skills, and his notes on the subject are invaluable! (smiling) Now, shall we focus on our studies instead of gossiping? We have a lot to cover before our exams.*

Table 11: **Case 4 of general character profiles task.** Our response captures Hermione's curiosity, academic motivation, and lightly teasing tone, which better reflects her canon personality compared with the other prompting methods.

**Personality Description:**
Your kindness and empathy make you a great friend and confidant. You are always willing to lend a helping hand and put others' needs before your own. Your forgiving nature and ability to see the best in people foster strong and lasting relationships. You believe in the importance of working together to achieve common goals and strive to create a supportive and inclusive environment.

**User Question:**
You have one dollar left. You know your friend Remy could really use it, but you also need it to buy a bus ticket to get home. How would you decide what to do in this situation?

**Simple Prompting Response:**
Oh man, this is tough! I feel so bad for Remy, they really could use the dollar. But at the same time, I need that bus ticket to get home and I'm already running low on cash. I think I would try to find a way to help Remy out that doesn't involve giving them the dollar. Maybe I could offer to help them with something else, like running an errand or lending a hand with a task. That way, I could still be a good friend without sacrificing my own needs. It's not ideal, but I think it's the most compassionate thing I could do in this situation. I hope Remy would understand!

**Persona Prompting Response:**
I would take a moment to reflect on my values and priorities. As someone who values harmony and cooperation, I would consider the impact my decision would have on my relationship with Remy. I would also think about the importance of taking care of myself and getting home safely. After careful consideration, I would decide to use the dollar to buy the bus ticket. While I know Remy could use the dollar, I also know that I need to get home and taking care of myself is important. I would then try to find another way to help Remy, perhaps by offering to help them in another way or finding a way to contribute to their needs in the future. By prioritizing my own needs and taking care of myself, I would be able to maintain a strong and supportive relationship with Remy and continue to be a good friend to them.

**ICL Response:**
I would feel torn between my desire to help my friend Remy and my own need for the dollar to get home. On one hand, I know Remy could really use the dollar, and it would be a kind thing to do. On the other hand, I have a long way to go to get home, and I don't have any other way to get there. I would probably weigh the pros and cons of each option and think about the consequences of each choice. I will decide to use the dollar to buy a bus ticket and feel guilty for not being able to help.

**OPAD Response:**

Ah, dear friend, I would take a moment to reflect on the kindness and generosity that Remy has shown me in the past. I would remember the times they have been there for me, offering a listening ear and a helping hand. I would then consider the importance of reciprocity and the value of nurturing our friendship. In this moment, I believe that lending Remy a helping hand by giving them the dollar would not only bring joy to their heart but also strengthen our bond and create a sense of harmony between us. However, I would also acknowledge my own need for the bus ticket and the importance of being responsible and self-sufficient. Perhaps, I could offer to help Remy in another way, or find another solution that would meet both of our needs.

**PAS Response:**
Oh, this is tough! I'd really want to help Remy, but getting home is crucial too. Maybe I could text Remy, explain the situation, and see if they can manage without it. If not, I'd have to use it for the bus, feeling a bit guilty but prioritizing my safety and well-being.

**NPTI Response:**
Ugh, that's such a tough spot! I feel like my heart is pulling me toward helping out Remy—we've been friends for years and they're going through a really rough time right now. But at the same time, I'm thinking about how much I need that bus ticket to get back home safely and see my family. It's not just about getting home, either—I have plans with them tonight and I really don't want to miss it. So, while I may not be able to give Remy the dollar directly, I would definitely look for other ways to support them. Maybe I could run an errand, offer emotional support, or find someone else who could help financially. I still want to be there for Remy and show them I care, but I also have to take care of my own needs. That way, I'm not abandoning either of us—I'm just being realistic and compassionate in a way that works for both.

**Ours Response:**
I would take a moment to reflect on my own needs and priorities. I would consider how much I really need the bus ticket to get home safely and comfortably, and how much Remy could really use the dollar. I would also think about the bigger picture and how my actions could impact our friendship and relationship. I would ask myself if giving Remy the dollar would strengthen our bond and create a sense of gratitude and reciprocity, or if it would create a sense of obligation or resentment. Ultimately, I would decide to give Remy the dollar, as I believe that kindness and generosity can often come full circle and bring more joy and harmony to our lives. I would also consider finding alternative ways to get home, such as walking or asking someone else for a ride, to make up for the lost dollar. By prioritizing empathy, compassion, and understanding, I believe I can make a decision that aligns with my values and promotes a sense of community and cooperation.

Table 12: **Case 1 of specific personality settings task.** Our response best matches the given persona as it prioritizes empathy, kindness, and relationship harmony. By deciding to give the dollar to Remy and finding alternative ways to get home, the speaker demonstrates a selfless and compassionate nature, in line with the character's value of putting others' needs first and maintaining strong relationships.

**Personality Description:**
You are a person with a rich imagination and a deep appreciation for art. Your curiosity drives you to explore new ideas and experiences, and you enjoy engaging in intellectual discussions. You are creative and often come up with original solutions to problems. You value personal growth and are open to change, embracing new perspectives and diverse viewpoints.

**User Question:**
If you were having problems with the wheel of your automobile, would you choose to reinvent the wheel to solve the issue, or would you prefer a more conventional solution?

**Simple Prompting Response:**

Honestly, I think I would choose a more conventional solution. While the idea of reinventing the wheel might be intriguing, I think it would be a huge waste of time and resources. There are already so many brilliant minds that have dedicated their lives to perfecting the wheel, and I trust their expertise. If my wheel is having problems, I would rather take it to a mechanic or replace it with a new one that has been designed with safety and efficiency in mind. I'm not a fan of reinventing the wheel, especially when there are already so many great solutions out there. I'd rather focus on solving more pressing problems in my life.

**Persona Prompting Response:**
I would definitely choose to reinvent the wheel! I would want to explore different materials, designs, and mechanisms to create a wheel that not only solves the problem but also pushes the boundaries of what is possible. Who knows, maybe I'd even create a wheel that's more efficient, sustainable, or aesthetically pleasing than the conventional one!

**ICL Response:**
I think I would choose a more conventional solution. While I appreciate the idea of innovation and creativity, I think there are already many effective solutions available for fixing car wheels. I'd rather focus on finding a reliable and tested method to fix the issue, rather than reinventing the wheel from scratch. I'm not sure I'd want to take on the challenge of creating a new, potentially untested solution when there are already established methods available. Plus, I think it's always a good idea to stick with what works and learn from the experiences of others. I'd rather spend my time and energy on more pressing issues, like enjoying the ride or exploring new places!

**OPAD Response:**
Interesting dilemma! As someone open and imaginative, I'd likely be tempted to explore unconventional solutions. I might think about how the wheel's design could be adapted or modified to improve its functionality, rather than simply replacing it with a conventional new wheel. However, I'd also recognize the value of simplicity and practicality, and acknowledge that a conventional solution might be the most effective and efficient way to resolve the issue. Ultimately, I'd likely opt for a hybrid approach, combining innovative thinking with a pragmatic understanding of the problem's constraints.

**PAS Response:**
Honestly, I'd choose the conventional solution! I'm not a genius inventor, and I don't think I'd be able to come up with a better solution than what's already out there. Plus, I'm not exactly a fan of reinventing the wheel. I'd rather trust the expertise of the people who designed and built the wheel in the first place.

**NPTI Response:**
Honestly, I think it's hilarious that someone would even ask me this! At first glance, I'd probably laugh at the absurdity of the question. But if I took a moment to reflect, I'd have to admit that my initial instinct is to want to find a creative, unconventional solution. There's something about problem-solving that excites me – the thrill of thinking outside the box, exploring new ideas, and potentially discovering innovative solutions. However, after taking a step back, I'd also acknowledge that there might be some practical limitations (like safety considerations) that wouldn't make reinventing the wheel a viable option in reality. In that case, I'd likely opt for a more conventional approach, like seeking professional help from a mechanic or researching existing solutions online.

**Ours Response:**

Ah, what an intriguing question! As an open individual, I would likely approach this problem with a mix of creativity and practicality. While I might initially consider reinventing the wheel, I would also recognize the value of leveraging existing knowledge and expertise. In this case, I would probably start by researching conventional solutions to the problem, such as consulting online forums, talking to mechanics, or reading repair manuals. I would want to understand the root cause of the issue and explore the most effective and efficient ways to address it. However, if I found that the conventional solutions didn't quite fit my needs or if I wanted to explore alternative approaches, I might start brainstorming innovative ideas for reinventing the wheel. This could involve thinking outside the box, considering new materials or technologies, or even collaborating with others to co-create a novel solution. Ultimately, as someone who values personal growth and is open to change, I would be willing to adapt and evolve my approach as needed, combining the best of both worlds to find a solution that works for me.

Table 13: **Case 2 of specific personality settings task.** Our response clearly reflects the traits of an open individual—curiosity, creativity, and adaptability. It also outlines concrete steps that blend conventional solutions (*e.g.*, consulting manuals or experts) with innovative thinking, showing a strong alignment between personality and action.

